# CD38 Enhances TLR9 Expression and Activates NLRP3 Inflammasome after Porcine Parvovirus Infection

**DOI:** 10.3390/v14061136

**Published:** 2022-05-25

**Authors:** Yi Zheng, Yixuan Xu, Weimin Xu, Sanjie Cao, Qigui Yan, Xiaobo Huang, Yiping Wen, Qin Zhao, Senyan Du, Yifei Lang, Shan Zhao, Rui Wu

**Affiliations:** Research Center of Swine Diseases, College of Veterinary Medicine, Sichuan Agricultural University, Chengdu 611130, China; zhengyi132@126.com (Y.Z.); xyxtk@foxmail.com (Y.X.); xuweimin10@163.com (W.X.); csanjie@sicau.edu.cn (S.C.); yanqigui@126.com (Q.Y.); rsghb110@126.com (X.H.); yueliang5189@163.com (Y.W.); zhao.qin@sicau.edu.cn (Q.Z.); senyandu@163.com (S.D.); y_langviro@163.com (Y.L.); zhaoshan419@163.com (S.Z.)

**Keywords:** CD38, PPV, Porcine Parvovirus, TLR9, IFN-α, NLRP3, CASP1, SIRT1

## Abstract

(1) Background: Porcine Parvovirus (PPV) is a single-stranded DNA virus without envelope which causes great harm in relation to porcine reproductive disorders in clinic. Cluster of Differentiation 38 (CD38) is a transmembrane protein widely existing in mammals. Its various functions make it a very popular research object, including in the viral infection field. (2) Methods: Western blotting and an EdU Cell Proliferation Kit were used to evaluate the effect of CD38-deficient cells. Relative quantitative real-time RT-PCR was used to detect the transcription levels of cytokines after PPV infection. The renilla luciferase reporter gene assay was used to verify the activation function of CD38 on downstream factors. The fluorescence probe method was used to detect the level of intracellular reactive oxygen species (ROS). (3) Results: This study found that the loss of CD38 function inhibited the up-regulated state of Toll-like Receptor 9 (TLR9), Interferon-α (IFN-α), and Myxovirus Resistance 1 (Mx1) after PPV infection. The luminescence of the group transfected with both CD38 expression plasmid and TLR9 promoter renilla luciferase reporter plasmid was significantly up-regulated compared with the control, suggesting that CD38 may activate the promoter of TLR9. In addition, CD38 deficiency not only activated the transcription of Sirtuin-1 (SIRT1), but also inhibited ROS level and the transcription of NLR Family Pyrin Domain Containing 3 (NLRP3). (4) Conclusion: (i) CD38 may participate in the TLR9/IFN-α/Mx1 pathway by activating the expression of TLR9 after PPV infected PK-15 cells; (ii) CD38 may activate the NLRP3/CASP1 pathway by increasing ROS level; (iii) CD38 deficiency activates the expression of SIRT1 and can prevent the normal proliferation of PPV.

## 1. Introduction

PPV is a member of *Parvovirus* in *Parvoviridae*. Porcine Parvovirus infection (PPI) has typical symptoms of stillbirth, mummification, embryonic death, and infertility (SMEDI), and is a reproductive disorder infectious disease that causes great losses to the global pig industry [1,2,3]. According to genetic differences, PPV is currently divided into seven genotypes. In addition to classical parvovirus type 1 (PPV1), PPV2–PPV7 have been successfully isolated and identified all over the world since 2001 [4,5]. As more and more new PPV strains are isolated and identified, even if with a few nucleotide mutations, their antigenicity and virulence may be very different, so it is necessary to further clarify the pathogenic mechanism of the virus.

The CD38 protein widely exists in peripheral blood, lymph nodes, and bone marrow. It is usually regarded as an activation marker of T cells and can be used as a monoclonal antibody target and for a variety of immunotherapy targets [6,7,8]. Besides immune cells, CD38 can also be expressed on other cells, but it is mainly concentrated in precursor cells and less in mature cells [6]. Therefore, the level of CD38 on the cell surface can assist in judging the stage of cell maturation and activation. In addition to being a transmembrane protein, CD38 can also be dissolved in body fluids to play its function [8]. Another important function of CD38 is its ability as nicotinamide adenine dinucleotide (NAD) hydrolase (NADase). At neutral pH, CD38 can hydrolyze NAD^+^ to adenosine diphosphate ribose (ADPR) and cyclic adenosine diphosphate ribose (cADPR), in which cADPR can also be hydrolyzed to ADPR. However, since the hydrolase activity is stronger than the cyclase activity, CD38 prefers to hydrolyze NAD^+^ to ADPR [9]. Besides, CD38 can convert nicotinic acid (NA) to nicotinic acid adenosine dinucleotide phosphate (NAADP) in an acidic pH environment [9]. All these products of CD38 are very efficient second messengers, which can cause the migration of Ca^2+^ [10]. Therefore, the expression of CD38 is related to the release of Ca^2+^, which can affect a variety of physiological functions such as cytokine production and vesicle transport. Accordingly, the expression of CD38 is also regulated by many factors, including TNF-α, IL-13, INFs, and other cytokines. Others, such as retinoic acid, lipopolysaccharide (LPS), and hormones also regulate CD38 [11]. For example, 17β-Estradiol can significantly increase the expression of CD38 in mouse airway smooth muscle cells. At this time, SIRT1 is down-regulated and p53 acetylation is increased, and these phenomena of SIRT1 and p53 disappear in CD38^−/−^ mice [12]. Another study [13] on CD38^−/−^ mice found that, after LPS stimulation, TLR4 expression in kidney tissue was up-regulated, the NF-κB pathway was activated, p65 migrated from cytoplasm to nucleus, and IFN-γ was up-regulated, too. At this time, sepsis was exacerbated. However, another study [14] showed that, under the environment of proinflammatory factors, CD38 and cADPR were up-regulated, resulting in an increase in Ca^2+^ levels, but it was not related to NF-κB and AP-1 pathways. Therefore, there is a further problem with the relationship between CD38 and NF-κB.

In conclusion, CD38 may have an indirect antiviral effect, and play a role by regulating immune response. The CD38/cADPR pathway has been proved to contribute to the activation of inflammatory response. The impairment of innate immune function in CD38^−/−^ mice also proves the importance of CD38 in immune regulation. It is known that the ssDNA of PPV can be recognized by TLR9 and activate the IFN pathway to induce the expression of antiviral protein. As a multifunctional protein, CD38 is regulated by a variety of regulatory factors including IFN. Therefore, it is of interest to know whether CD38 affects the TLR9-dependent IFN pathway during PPV infection, whether it has an impact on the ROS-related NLRP3/CASP1 pathway, and whether there are other cytokines involved in resisting virus infection after CD38 deficiency.

## 2. Materials and Methods

### 2.1. Virus and Cells

HNLY201301 strain (Accession no. MF447833), which was previously analyzed and uploaded by our laboratory, is a classical genotype 1 porcine parvovirus strain isolated from Henan Province, China in 2013. It was passaged in PK-15 cells (ATCC), and virus titer was determined by TCID50 assay. HEK-293T cells (ATCC), PK-15 cells, and CD38 knockout PK-15 cells (KO.CD38 cells, generated from PK-15 cells as parental cells) were maintained in Dulbecco’s modified eagle medium (DMEM) (Gibco, Grand Island, NY, USA) supplemented with 10% fetal bovine serum (FBS; Hyclone, South Logan, UT, USA), 100 μg/mL streptomycin, and 100 U/mL penicillin in a 5% CO_2_ incubator at 37 °C.

### 2.2. Plasmids and Antibodies

The full-length *CD38* gene (Accession no. NM_001243883) was amplified from PK-15 cells by PCR. The pCMV-HA-CD38 plasmid was constructed by inserting the full-length PCR-amplified *CD38* gene into the pCMV-HA vector, which was digested with Xho I and Not I. The promoter sequences of TLR9 and Mx1 genes with a length of about 2000 bp were amplified from PK-15 cells by PCR. The pGL-TLR9p and pGL-Mx1p plasmids were constructed by inserting the PCR-amplified promoter into the pGL4.79 vector, which was digested with Xho I and Hind Ⅲ. The primers used for plasmids construction and quantitative real-time RT-PCR in this study are listed in Table 1. Sequences of all the recombinant plasmids were verified by Sanger dideoxy sequencing. The rabbit polyclonal anti-CD38 antibody was purchased from Thermo Fisher Scientific (Waltham, MA, USA) and the rabbit monoclonal anti-β-actin antibody was supplied from ABclonal Technology (Wuhan, China).

### 2.3. Virus Infection and Copy Number Detection

If PPV is infected on the passage cells too late, or too few virus particles are infected, the cytopathic effect is not obvious. Therefore, this study adopts the method that the virus is added at the same time of cell passage, which means cells are infected with PPV when they are suspended, and then continues to culture in 37 °C, 5% CO_2_ atmosphere for the indicated time. The mock-infected cells were prepared using the same procedures without virus.

The copy number of PPV was detected by quantitative PCR, and the primers sequences were 5′-GGGCTTGGTTAGAATCACT-3′, R: 5′-ACACTCCCCATGCGTTAG-3′. This method is based on the structural protein VP2 of PPV.

### 2.4. Construction of CD38 Knockout PK-15 Cell Line by CRISPR/Cas9 System

The online CRISPR tool (http://crispr.mit.edu/, accessed date: 17 September 2019) was used to design specific sgRNA for CD38 (sgRNA-F: 5′-CACCGCGTCGTATCTGGATCCGAGG-3′ and sgRNA-R: 5′-AAACCCTCGGATCCAGATACGACGC-3′). The two oligos of sgRNA were cloned into LentiCRISPR-V2 plasmid (Addgene). Next, the recombinant plasmids psPAX (Addgene) and pMD2.G (Addgene) were co-transfected into HEK-293T cells using Lipofectamine 3000 (Invitrogen, Carlsbad, CA, USA). The supernatant at 24 h after transfection was filtered and collected as lentivirus solution harboring the target sgRNA for infecting PK-15 cells. Puromycin (5.5 μg/mL) was added for selecting positive cells. Cell clones with CD38 knockout were confirmed by DNA sequencing and Western blotting. Meanwhile, CD38.KO cells were transfected with the pCMV-HA-CD38 plasmid to evaluate the effect of trans-complementation.

### 2.5. Renilla Luciferase Reporter Gene Assay

Renilla luciferase is a 36 kDa protein which can immediately catalyze the enzyme-independent oxidative luminescence of coelenterazine and play a role as a reporter gene. The experiment was carried out in a 6-well plate, pGL-TLR9p and pGL-Mx1p plasmids were co-transfected with pCMV-HA-CD38 plasmid into HEK-293T cells, and the group transfected with reporter plasmid alone was used as control. At 36 h after transfection, the cells were gently washed with PBS. After adding lysate into each well, cells were immediately scraped off with a cell scraper and pipet repeatedly to ensure the harvest of homogeneous suspension. After two freeze–thaw cycles, the suspension was centrifuged at 13,000 r/min, 4 °C for 30 s to obtain supernatant. The reaction solution containing coelenterazine and the supernatant were quickly mixed in a dark environment, and then detected immediately. Luminescence was set to detect for integrated over 10 s with a 2 s delay.

### 2.6. Reactive Oxygen Species Assay

In order to detect ROS level, a DCFH-DA probe was loaded into the cell, then oxidized by ROS, and the fluorescence signal was captured by the instrument. The cells to be tested were cultured in a 96-well plate (10^4^ cells per well) in a 37 °C and 5 % CO_2_ incubator, with 3 replicates in each group. Cells were infected with PPV at multiplicity of infection (MOI) of 1, ROS level was detected at 24 hpi (hours post-infection), and the mock-infected cells were prepared using the same procedures without virus. The DCFH-DA probe was diluted to 10 μmol/L by serum-free DMEM and then added to a 96-well plate (100 μL per well), continuing to culture for 20 min. After being washed by serum-free DMEM, the fluorescence signal of the sample was detected immediately at 488 nm excitation wavelength and 525 nm emission wavelength.

### 2.7. EdU Cell Proliferation Assay

EdU is a novel thymidine analogue which can replace thymidine and be incorporated into the newly synthesized DNA. Through the Click Reaction between the acetylene group on EdU and the fluorescent-labeled small molecule azide probe, the newly synthesized DNA is fluorescent-labeled, and then the cell proliferation can be observed under the fluorescence microscope (BX53, OLYMPUS, Tokyo, Japan). BeyoClick™ EdU Cell Proliferation Kit with Alexa Fluor 555 (Beyotime Biotechnology, Shanghai, China) was used to detect cell proliferation in this study. All procedures were performed according to the manufacturer’s recommendations.

### 2.8. RNA Extraction and Quantitative Real-Time RT-PCR

Cells were treated or infected as indicated. Then, total RNA was extracted using TRIzol Reagent (Sangon Biotech, Shanghai, China). Reverse transcription was performed using PrimeScript™ RT reagent Kit with gDNA Eraser (Perfect Real Time) (TaKaRa Bio, Tokyo, Japan). Synthetic cDNA was analyzed for quantitative real-time PCR using TB Green^®^ Premix Ex Taq™ II (Tli RNaseH Plus) (TaKaRa Bio) in a LightCycler^®^ 96 System (Roche, Basel, Switzerland). Relative mRNA values were calculated using the 2^−∆∆CT^ method. β-actin was used as an internal control in each sample and showed as fold change by normalizing to the mock-control.

### 2.9. Quantification and Statistical Analysis

All statistical analyses and calculations were performed using GraphPad Prism (Version 7, La Jolla). All data are expressed as means ± standard deviation (SD) as indicated. Student’s unpaired t-test was used to estimate the statistical significance between two groups, whereas ANOVA was used to compare the means among three or more groups. A p-value less than 0.05 was considered statistically significant. Statistical significance is indicated as * (*p* < 0.05), ** (*p* < 0.01), *** (*p* < 0.001), and ns (not significant).

## 3. Results

### 3.1. CD38.KO Cell Line Was Constructed and Proliferated Normally

CD38.KO cell lines were constructed using the CRISPR/Cas 9 system, and cells with frameshift near protospacer adjacent motif (PAM) sequence were successfully screened (Figure 1A). Although a base is inserted upstream of the PAM sequence, it is not clear whether the synthesis of the target protein is affected. In order to verify the expression of CD38 protein in the CD38.KO cells, Western blotting was used in this study. At the same time, the expression effect of CD38 was also detected after the CD38.KO cells were transfected with pCMV-HA-CD38 plasmid. As shown in Figure 1B, CD38 was not expressed in CD38.KO cells, but could be expressed normally after pCMV-HA-CD38 plasmid transfection. The proliferation of CD38.KO cells was detected by EdU assay. The newly synthesized DNA labeled by Alexa fluor 555 shows red fluorescence, and Hoechst 33342, a common nuclear stain, shows blue fluorescence. When CD38.KO cells and PK-15 cells were cultured under the same conditions for 2 h, there was no significant difference in the proportion of EdU-positive cells, indicating that the defect of CD38 gene had no effect on cell proliferation (Figure 1C).

### 3.2. CD38 Participates in the TLR9/IFN-α/Mx1 Pathway Activated by PPV

Samples of 24 hpi were used for quantitative real-time RT-PCR detection. A variety of cytokines including TLR9, RIG1, IRF1, IRF3, IRF7, IFN-α, IFN-β, TNF, IL6, IFIT1, and Mx1 were analyzed. The amplification efficiency of the primers used was between 90 and 110%. The results showed that the transcriptional levels of TLR9, IFN-α, and Mx1 were up-regulated after PK-15 was infected by PPV, and these up-regulated phenomena disappeared when CD38 gene was deleted (Figure 2A). In addition, there is no such phenomenon in other detected cytokines (data not shown). The relative mRNA levels of IFN-α and CD38 in PK-15 cells were analyzed at multiple post-infection times. As shown in Figure 2B, the transcriptional trend of IFN-α and CD38 both gradually increased and then decreased to the pre-exposure level at 48 hpi. Moreover, the results of comparing the relative mRNA level of IFN-α in PK-15 cells and CD38.KO cells are shown in Figure 2C. There was no fluctuation in CD38.KO cells, suggesting that CD38 may be closely related to the expression of IFN-α. The results of renilla luciferase reporter gene assay are shown in Figure 2D. Only the samples co-transfected with pCMV-HA-CD38 plasmid and pGL-TLR9p plasmid have significantly increased luminescence signal intensity compared with the control.

### 3.3. Loss of CD38 Function Inhibited ROS/NLRP3/CASP1 Pathway Activated by PPV

The level of intracellular ROS was detected by probe method. The results are shown in Figure 3A. After PPV infection, the level of ROS in PK-15 cells was significantly up-regulated compared with that in mock-infected cells, and the up-regulation of ROS was inhibited when CD38 gene was deleted. The transcription levels of NLRP3 and CASP1 were detected by the same method as before, and the results are shown in Figure 3B. The relative mRNA level of NLRP3 was up-regulated by at least 50-fold in PK-15 cells compared with the control, while it was down-regulated in CD38.KO cells. At the same time, the relative mRNA level of CASP1 was up-regulated at least 2-fold in PK-15 cells and significantly up-regulated at least 100-fold in CD38.KO cells.

### 3.4. SIRT1 Is Inhibited by CD38 and Is Conducive to PPV Infection

The quantitative real-time RT-PCR results of SIRT1 transcription level are shown in Figure 4A. Comparing the two kinds of cells not infected with PPV, the relative mRNA level of SIRT1 was significantly up-regulated in CD38.KO cells than in PK-15 cells. Meanwhile, the transcriptional level of SIRT1 was differentially down-regulated after PPV infection (24 hpi) in PK-15 cells, but the opposite phenomenon was observed in CD38.KO cells. In other words, the transcription of SIRT1 is inhibited in PPV-infected PK-15 cells, while CD38 deficiency makes it up-regulated and more significant. Nicotinamide (NAM) is one of the commonly used inhibitors of SIRT1. In order to determine the appropriate working concentration of NAM, it was diluted into different concentrations by PBS and incubated with CD38 cells, and the EdU assay results are shown in Figure 4B. The proportion of EdU-positive cells in samples with NAM concentrations of 5 mM, 25 mM, and 50 mM had no significant difference with the control, that is, the normal proliferation of CD38.KO cells was not affected by these three concentrations of NAM. Then, the relative mRNA level of SIRT1 in CD38.KO cells treated with different concentrations of NAM and infected with PPV (24 hpi) was detected. As shown in Figure 4C, the inhibition of SIRT1 by NAM was statistically different from that in the control, and the higher the concentration, the better the inhibition effect. At this time, the PPV copy number was up-regulated in a dose-dependent manner (Figure 4D).

## 4. Discussion

In our previous study, the Whole Genome sgRNA Library of pigs was constructed by PK-15 cells and was applied to screen the key genes of PPV-infected host cells. After the analysis, CD38 was obtained with a high score (unpublished data). Innate immunity is the first line of host for defending the invasion of pathogen. Type I IFN, including IFN-α and IFN-β, represents the main response of the innate immune system to virus invasion [15]. The signal pathway of type I IFN is activated and expressed after the viral component or replication intermediate is recognized by pattern recognition receptor (PRR) [16]. Subsequently, the secreted type I IFN binds to IFN-α/β Receptor and initiates subordinate signal transduction through the JAK/STAT pathway [17]. It then activates the transcription of multiple IFN-stimulated genes (ISG) involved in antiviral immunity, such as dsRNA-dependent protein kinase (PKR), 2′–5′ oligoadenylate synthetase (OAS), Mx1, ISG15, ISG56, and ISG56 [18]. We described the results of PPV-stimulated PK-15 cells for 24 hpi, which show that the transcriptional levels of TLR9, IFN-α, and Mx1 were up-regulated. In other words, after PPV internalization into cells, it is recognized by TLR9 as PRR in the endosome, then stimulates IFN-α expression, and finally Mx1 plays an antiviral role. The ssDNA of the parvovirus family is mainly recognized by TLR9, such as Minute virus of mice (MVM), Adeno-associated virus type 2 (AAV2), and Rat parvovirus KRV strain [19,20]. Our findings are consistent with previous research showing that the ssDNA of PPV is also recognized by TLR9. Transcription of type I IFN requires activation of IRF3, ATF2-c-Jun, and NF-κB, but the nonstructural protein NS2 of PPV is an antagonist of IFN, that is, PPV may inhibit the activation of other transcription factors such as IRF3 and ATF2-c-Jun, thus blocking the expression of IFN gene [1], which may explain why we did not detect the significant up-regulation of other important factors on IFN pathway at 24 hpi such as IFN-β and IRF3 (data not shown).

Many studies have shown that increased expression of CD38 is one of the markers activated by a variety of inflammatory factors, and the close relationship between type I IFN and CD38 gene has been found in epithelial cells, renal cells, macrophages, and many other cells [11]. For instance, oxidative stress and the expression of IFNs, ISGs, and CD38 basically occurred synchronously in the process of respiratory syncytial virus (RSV) infection, that is, CD38 enhanced the inflammatory response after RSV infection. When the flavonoids drug Kuromanin was used to inhibit the ADPR cyclase activity of CD38, and the cADPR analogue 8-br-cADPR was used to antagonize the function of cADPR, the inflammatory response induced by RSV in cells was significantly reduced [21]. In order to avoid a cytokine storm caused by excessive positive regulation of inflammatory response by CD38, other proteins such as tristetraprolin (TTP) [22] are usually induced in the regression stage of inflammation to prevent the pro-inflammatory function of CD38. Our research confirmed this viewpoint by finding that the transcriptional of CD38 was down-regulated to the normal level after 48 hpi. The up-regulation of CD38 mRNA was the most significant at 24 hpi, which is the reason why we mainly chose this time point for the experiments. However, some studies have found that there is a continuous expression of CD38 in CD8^+^ T cells after Human parvovirus B19 (HPVB19) infection, which may contribute to the long-term control of HPVB19 [23]. In addition, it has been reported that a variety of cytokines in CD38^−/−^ mice, such as TNF-α, iNOS, IL6, IL1β, CCL2, CCL3, CXCL10, CD68, and TREM2, are no longer up-regulated after pathogen stimulation [24]. In line with previous studies, we found that TLR9, IFN-α, and Mx1, which should be up-regulated, were inhibited in CD38.KO cells. The subsequent renilla luciferase reporter gene assay found that CD38 protein can activate the promoter of TLR9. From this result, CD38 can be considered as the factor contributing to the expression of TLR9 after PPV infection. It is worth noting that the luminescence intensity is positively correlated with the expression of luciferase. During co-transfection assay, only half of the transfected plasmids contain luciferase gene compared with the control group. Therefore, when the promoter in the reporting plasmids is not activated, their luminescence intensity may be less than half of the control group.

Inflammasome is a protein complex that can recruit and activate CASP1, then shear, and trigger the secretion of proinflammatory factors such as IL-1β, and induce programmed cell death [25]. This study found that the loss of CD38 function led to the inhibition of the up-regulation of ROS after PPV stimulation, which made the NLRP3 inflammasome closely related to the release of Ca^2+^ and ROS become the object of further research. A conclusion was reached by quantitative experiments. PPV can activate the NLRP3/CASP1 pathway and start the process of apoptosis. However, after the lack of CD38, the transcription level of NLRP3 changes in the opposite way. Obviously, after the loss of CD38 function, NLRP3 inflammasome cannot be activated. Under some assumptions, this can be explained in that CD38 deficiency affects the release of intracellular Ca^2+^ and inhibits the up-regulation of ROS, resulting in the inability of the NLRP3/CASP1 pathway to activate. In addition, the transcription level of CASP1 in CD38.KO cells was significantly up-regulated after PPV infection. We speculate that this might be due to the transcription of CASP1 induced by pathogen stimulation, but it cannot be recruited through NLRP3 to produce effector protein. On the one hand, precursor CASP1 accumulates in cells, and on the other hand, CASP 1 may also be recruited through other inflammasome pathways, such as AIM2 and IFI16. As NADase, CD38 has very significant catalytic efficiency. Therefore, CD38 is an important regulator of the intracellular NAD^+^ pool and a variety of metabolic pathways. Due to its very significant NADase catalytic efficiency [12], CD38 is an important regulator of the intracellular NAD^+^ pool and a variety of metabolic pathways. SIRT1 also takes NAD^+^ as substrate, and some virus infections can induce apoptosis and inflammation by promoting ROS production and reducing SIRT1 expression [26,27]. Therefore, the up-regulation of SIRT1 can hinder virus proliferation by inhibiting apoptosis and inflammation. The activity of SIRT1 increases when NAD^+^ is at a high level, but when CD38 is abundant, the activity of SIRT1 decreases because a large amount of NAD^+^ is consumed [28]. As discussed, we consider that when CD38 is lacking, SIRT1 is activated in a high-level NAD^+^ environment, and then inhibits apoptosis by weakening ROS response, so as to hinder the normal proliferation process of PPV to protect cells. It is important to note that the present evidence relies on the study of transcription level, and the study of protein expression level is the key to the next step.

## 5. Conclusions

Taken together, this study demonstrated two things. (i) The process of PPV invading PK-15 cells is that its viral ssDNA is first recognized by TLR9 on the endosome, activates the IFN pathway, synthesizes and secretes Mx1 and other antiviral proteins, and the host protein CD38 can participate in this process by enhancing the expression of TLR9. (ii) CD38, as a kind of NADase, on the one hand inhibits the transcription of SIRT1, and on the other hand promotes the release of intracellular Ca^2+^ and up-regulates the level of ROS through CD38/cADPR pathway. At this time, NLRP3 inflammasomes are activated, and CASP1 can activate inflammatory response and programmed cell death (Figure 5).

## Figures and Tables

**Figure 1 viruses-14-01136-f001:**
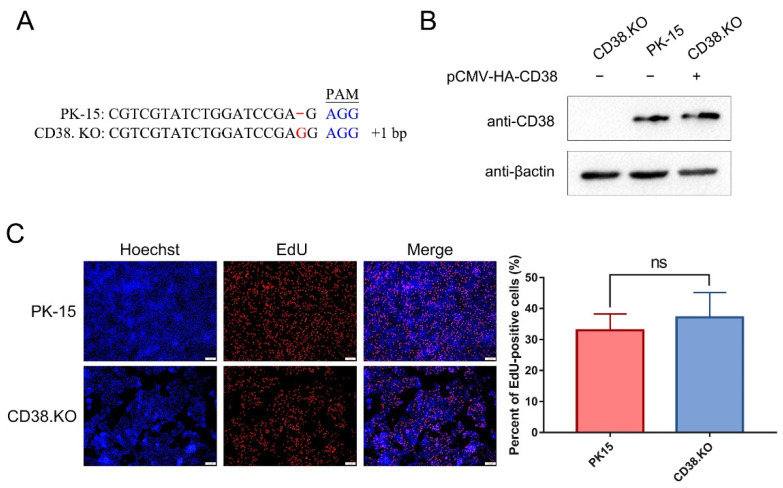
CD38.KO cell line was constructed and proliferated normally. (**A**) Sequence alignment between CD38.KO cells and PK-15 cells. The red font represents the sequence could not be paired, and the blue font represents the PAM sequence. (**B**) Western blotting was used to verify the expression of CD38 in CD38.KO cells, PK-15 cells, and CD38.KO cells transfected with pCMV-HA-CD38 plasmid. (**C**) PK-15 cells and CD38.KO cells were cultured for 2 h after loading EdU. After the click reaction, photos were taken with fluorescence microscope (400×), EdU-positive cells are shown in red and nuclei are shown in blue. At least 3 photos of each sample were taken, the percentages of EdU-positive cells were calculated, respectively, and the average value was taken as the final data. ns: not significant.

**Figure 2 viruses-14-01136-f002:**
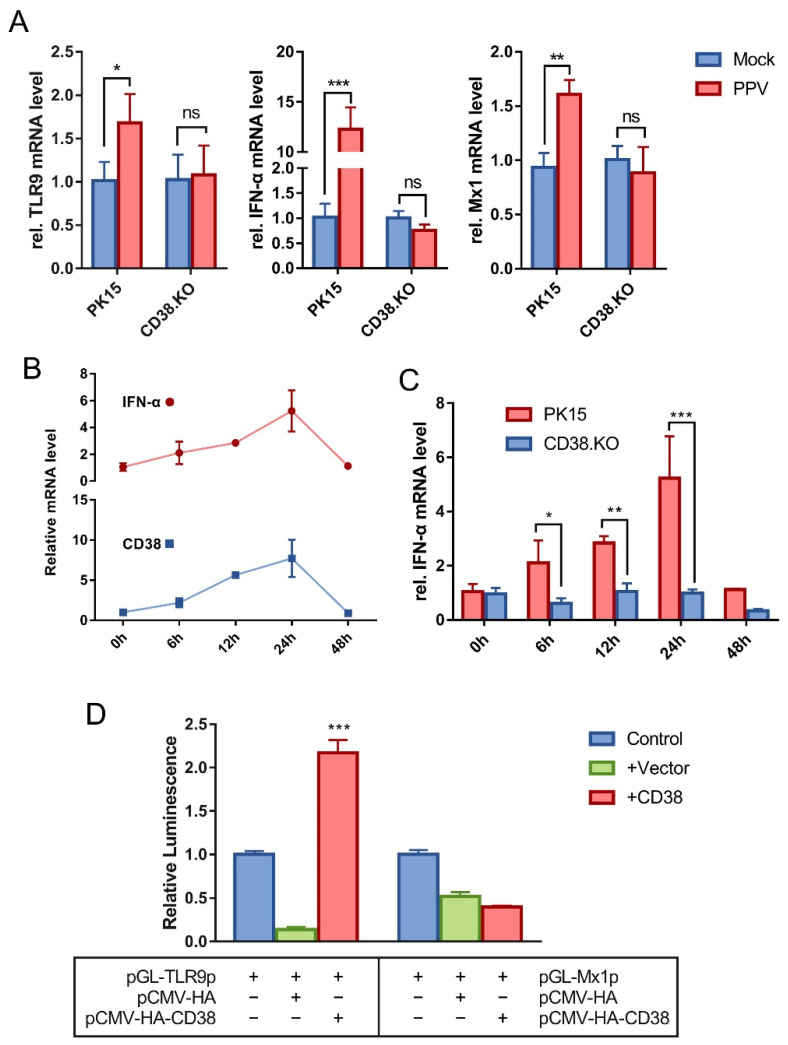
CD38 participates in the TLR9/IFN-α/Mx1 pathway activated by PPV. Quantitative real-time RT-PCR was used to detect the relative transcription level of cytokines in PPV-infected cells (MOI = 1), with β-actin as the internal reference gene. (**A**) The relative mRNA levels of TLR9, IFN-α, and Mx1 in CD38.KO cells (24 hpi), PK-15 cells as control. (**B**) The relative IFN-α and CD38 mRNA levels in PK-15 cells (0–48 hpi). (**C**) The mRNA level of relative IFN-α in CD38.KO cells (0–48 hpi), PK-15 cells as control. (**D**) The reporter plasmid containing promoter and the target plasmid were transfected into HEK-293T cells. The luminescence of Renilla luciferase detected is shown as fold change by normalizing to the sample-transfected-only reporter plasmid. *: *p* < 0.05, **: *p* < 0.01, ***: *p* < 0.001, and ns: not significant.

**Figure 3 viruses-14-01136-f003:**
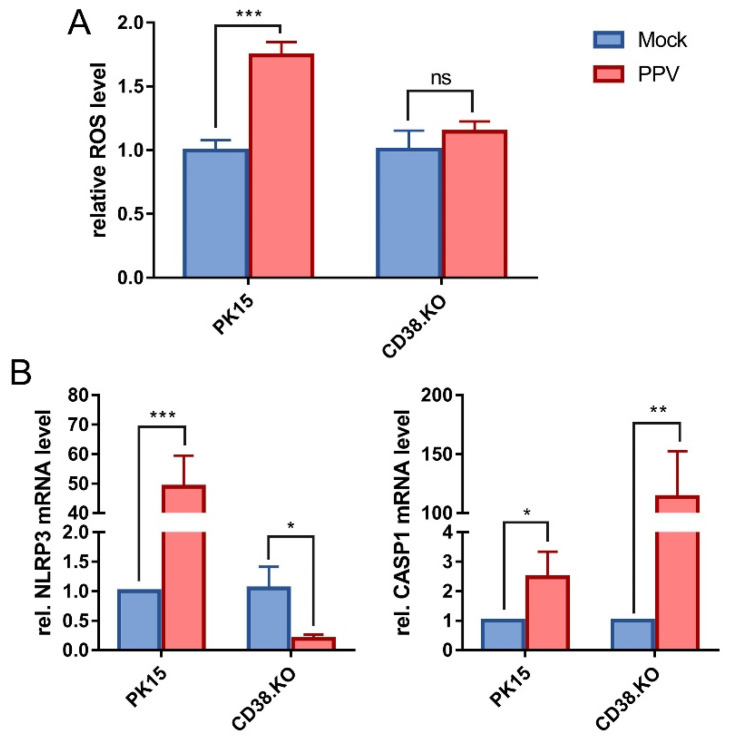
Loss of CD38 function inhibited ROS/NLRP3/CASP1 pathway activated by PPV. (**A**) The DCFH-DA probe was loaded into the cells infected with PPV (MOI = 1, 24 hpi). Fluorescence signals at 488 nm excitation and 525 nm emission wavelength were collected, and the relative level of intracellular ROS was calculated and is shown as fold change by normalizing to the mock-infected cells. (**B**) Quantitative real-time RT-PCR was used to detect the relative mRNA levels of NLRP3 and CASP1 in CD38.KO cells after PPV infection (MOI = 1, 24 hpi), and PK-15 cells were used as control. *: *p* < 0.05, **: *p* < 0.01, ***: *p* < 0.001, and ns: not significant.

**Figure 4 viruses-14-01136-f004:**
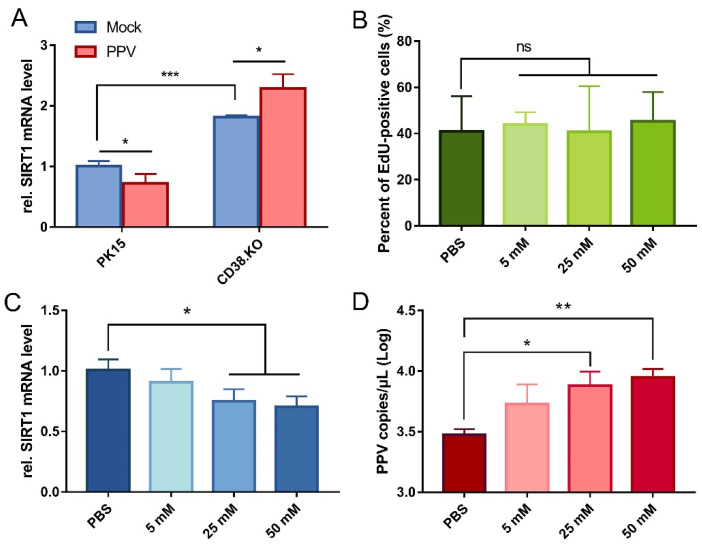
SIRT1 is inhibited by CD38 and is conducive to PPV infection. (**A**) Quantitative real-time RT-PCR was used to detect relative SIRT1 mRNA level in CD38 cells after PPV infection (MOI = 1, 24 hpi), and PK-15 cells were used as control. (**B**) CD38.KO cells treated with different concentrations of NAM continue to be cultured for 2 h after loading EdU. After the click reaction, at least 3 photos of each sample were taken by fluorescence microscope. The percentage of EdU-positive cells were calculated, respectively, the average value was taken as the final data, and the cells treated with PBS as the control. CD38.KO cells treated with different concentrations of NAM were infected with PPV (MOI = 1, 24 hpi). The mRNA level of SIRT1 (**C**) and the copy number of PPV virus (**D**) in the cells were detected. The cells treated with PBS were used as the control. *: *p* < 0.05, **: *p* < 0.01, ***: *p* < 0.001, and ns: not significant.

**Figure 5 viruses-14-01136-f005:**
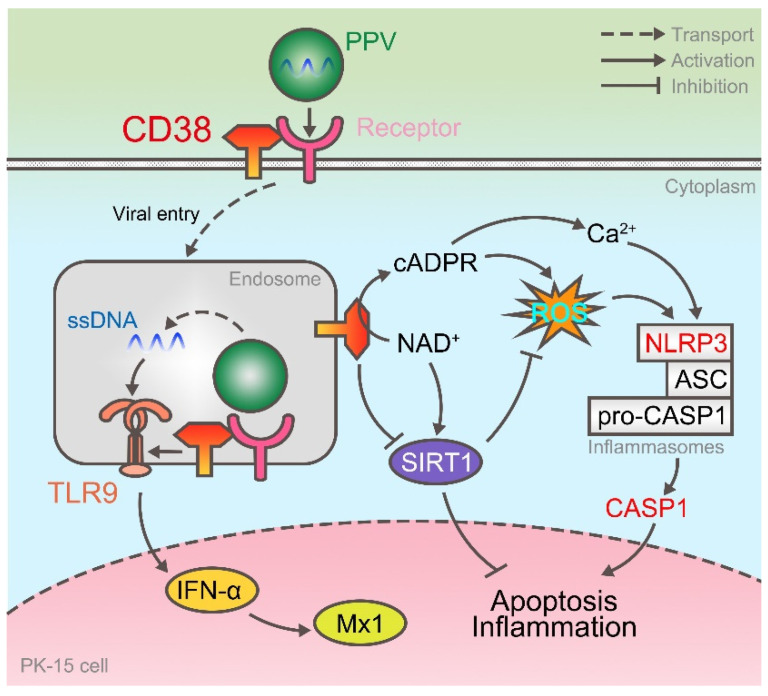
CD38 enhances TLR9 expression and activates NLRP3 inflammasome after PPV infection. On the one hand, after PPV enters the cell, its ssDNA is recognized by TLR9 and activates the expression of IFN-α and Mx1, and CD38 participates in it as the upstream of TLR9. On the other hand, CD38 enhances ROS levels and activates NLRP3 inflammasome by cyclizing ADPR and inhibiting SIRT1 expression.

**Table 1 viruses-14-01136-t001:** Primer pairs used for plasmid construction and quantitative real-time RT-PCR.

Primer Name	Forward (5′ → 3′)	Reverse (5′ → 3′)
HA-CD38	ggtcgaccgagatctctcgagGCCACCATGGCCAACCACAGATTCA	catgtctggatccccgcggccgcCTAGATAGGCCTGTAGTTTTCCTGG
GL-TLR9p	cctgagctcgctagcctcgagGGTGATTTGGAGGTAATTGTCA	cagtaccggattgccaagcttCCCCATCTCCTCCCCAGC
GL-Mx1p	cctgagctcgctagcctcgagCTCCTGCCAGATCAGTAAACG	cagtaccggattgccaagcttGCCCCCTGCCCTGAGCCC
Q-TLR9	TGGTTACCTGGCAAGACGC	GGAAACTGGCACGCAAGAG
Q-IFN	GGACTCCATCCTGGCTGTGA	GACTTCTGCCCTGATGATCTCC
Q-Mx1	CTGCCCCTGTTAGAAAACCAAA	GCCGACACTCGTACTCCAC
Q-NLRP3	GGTGAAGCGTTTGTTGAG	AATGGATGGGTTTACTGG
Q-CASP1	GTCCGAAGCGGTGAGATT	CCCAGACATAGCCCAAAG
Q-SIRT1	AGGGAACCTTTGCCTCACC	TGGCATATTCACCTCCTAACCTA
Q-βactin	CTTCCTGGGCATGGAGTCC	GGCGCGATGATCTTGATCTTC

## Data Availability

All data generated and analyzed in this research are included in the article.

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
