# Peer review of "CD38 Enhances TLR9 Expression and Activates NLRP3 Inflammasome after Porcine Parvovirus Infection"

_viruses, 2022, doi:10.3390/v14061136_

Round 1

Reviewer 1 Report

I generally think this is a good study, and a well-written manuscript. I don't have any comment.

Author Response

Thank you for your comments. Please review our revised version.

Reviewer 2 Report

This is a relevant study performed on CD38 during the Porcine Parvo Virus infection. The authors provided novel details of CD38 role in virus infection. For example, the TLR9-dependent IFN pathway and the ROS-related NLRP3/CASP1 (inflammasome) pathway are involved in the resistance of Porcine Parvo Virus infection.

Material and Method

1. In the virus infection section, the author mentioned adding viruses at the same time as passage cells, meaning the author infects the cells when they are in suspension or infect with a certain percentage of the cell monolayer? please clear this

2. In section 2.4,  please add at in this sentence supernatant 24 h

Results section in 3.1A please mention the name of cell line PK15 instead of normal. So it will be clear to the reader 

Reviewer 3 Report

Accept in present form. The results bring important contributions to the understanding of the pathogenicity of PPV.

Author Response

(The authors gave the same response as above.)

Reviewer 4 Report

Zheng et al studied the role of CD38 after infection with Porcine parvovirus (PPV) in PK-15 cells by generating CD38 knockout cells. Authors found that the loss of CD38 in PK-15 cells inhibited upregulation of TLR9, IFN-α, Mx-1 genes upon PPV infection. They also observed activation of Sirtulin-1 and inhibition of NLRP3 at transcript level. The work is interesting. However, there are weaknesses associated with this work.

Major comments

  1. Although authors mention throughout the manuscript as CD38.KO cell line, CD38 expression was only downregulated in these cells, and these cells still express about 25% of CD38 compared to PK-15 parent cell line.
  2. The conclusions of this study are based on expression at transcript level and not at protein level. Although authors mentioned this point in discussion, it is a major limitation of this study. Except for IFN-a and CD38, other cytokine and IFN gens were analyzed at single time point of 24 hpi. 
  3. It is not clear what is the rationale for using luciferase reporter system to assess TLR9 expression in transfected HEK293 cells instead of directly detecting expression levels of TLR9 by western blotting in PK-15 and CD38.KO cells.
  4. In Fig. 2D, effect of overexpression of CD38 was only observed for TLR9 reporter, but not for Mx1. This is inconsistent with the Fig. 2A, where they observed upregulation of Mx1 mRNA in CD38 intact PK-15 cells after PPV infection. Authors must discuss these differences in discussion section. Also, why was there a downregulation of luciferase signal in vector transfected cells?
  5. Authors should have used CD38 gain of function in CD38.KO cells after transfecting these cells with pCMV-HA-CD38 especially for the Fig 3 and 4 to support their conclusion.
  6. Is there a difference in PPV virus copy number in infected PK-15 and CD38.KO cell line?

Minor comments:

  1. Section 2.6 ROS assay it is mentioned that the PPV was infected at MOI of 100. It should be cells were infected with PPV at MOI of…… Also, why was MOI 100 used for this experiment? In figure legend it says MOI of 1.
  2. Method for PPV copy number assessment need to be included

Round 2

Reviewer 4 Report

I am happy with the response from authors.